# What if Employees Brought Their Life to Work? The Relation of Life Satisfaction and Work Engagement

**Pedro Ferreira [1,]*** , **Carla Gabriel [2]** , **Sílvia Faria [1]** , **Pedro Rodrigues [3]** and
**Manuel Sousa Pereira [4]**

1   Department of Economics and Management, University Portucalense, 4200-072 Porto, Portugal;
    sfaria@upt.pt
2   Higher InstituteMiguel Torga, 3000-132 Coimbra, Portugal; carlagabriel@ismt.pt
3   Department of Economics and Management, University Lusíada-North, 4369-006 Porto, Portugal;
    pmrodrigues@gmail.com
4   Polytechnic Institute of Viana do Castelo, 4900-347 Viana do Castelo, Portugal; msousa.manuel@gmail.com
*   Correspondence: pferreira@upt.pt

**Abstract:** Previous research on work engagement has sought to understand the drivers, as well as the consequences, of engaging people at work. The drivers of engagement have been found mainly within the work domain. However, working life is not detached from personal life, which has been demonstrated by research on work-life balance. The goal of this research is to understand the relation between life satisfaction and work engagement. Data were collected from a sample of 571 employees from major Portuguese companies using a questionnaire. The results confirmed the constructs used, and the regression model showed that life satisfaction is related to work engagement. The main conclusion was that work engagement can be explained by factors external to the organization, which are an integral part of employees' lives. Being a psychological and emotional state, work engagement can be related to other aspects of employees' lives besides their organizational and job roles.

**Keywords:** work engagement; life satisfaction; employee wellbeing; Utrecht Work Engagement Scale

---

## 1. Introduction

Sustainability is considered to be the next step in the Human Resource Management (HRM) evolution [1]. According to several authors [1,2], the HRM function is in a good position to help companies to be more socially responsible and sustainable. Following this perspective, several studies have sought to shed some light on the links between corporate social responsibility, sustainability and HRM [3–6]. Since sustainability seems to be closely related to the knowledge, skills and engagement that the workforce might bring to the attainment of sustainable goals [7], the investment and attention given to sustainable HRM policies and practices may be considered an important management decision.

Sustainable HRM can be understood as the set of policies and practices that enhance economic, environmental and social results, including employees, their families and their communities [8,9]. In a recently published systematic literature review about the connections between Corporate Social Responsibility, Sustainability and HRM, Podgorodnichenko, Edgar and McAndrew [2] found that among the most important issues found in the literature, HRM policies and practices should address employees' wellbeing and work-life balance [10,11], in order to achieve a sustainable workplace. This evidence supports the argument that employee wellbeing is closely related to sustainability. In fact, research conducted by Järlström et al. [12] came to highlight four dimensions of sustainable HRM: justice and equality; transparent HR practices; profitability; and employee wellbeing. Also, a recent

theoretical paper proposing a set of sustainable HRM characteristics included "care of employees", which included work-life balance as a core aspect [13].

Wellbeing can be considered as deriving from a general sense of physical and mental health [14]. Rothmann [15], based on research with members of the South African police force, suggests that work engagement is an important component of work-related wellbeing, alongside job satisfaction, occupational stress and burnout. In fact, a stream of the literature conceptualizes work engagement as the antipode of burnout [16], underlying the positive characteristics of work engagement as an important part of employees' wellbeing.

Previous research on work engagement has sought to understand the drivers, as well as the consequences, of engaging people at work. Among the most studied consequences, employee engagement seems to influence job satisfaction [17], organizational commitment [18], intention to quit, organizational citizenship behavior [19] and performance [18,20,21]. The drivers of engagement have been found mainly inside the organizational context, such as job resources and characteristics [21,22], meaningful work [23], organizational support, rewards and recognition and justice [24].

However, working life is not detached from personal life, which has been demonstrated by research on work-life balance [25]. Since work engagement is a psychological phenomenon characterized by a positive, fulfilling, affective-motivational state of work-related wellbeing [26], constraints outside the sphere of work may impact this psychological state. This potential relation calls for the need to examine the connection between employee wellbeing and engagement.

Satisfaction with life is within the realm of subjective wellbeing. This subject has become popular in recent years, and much research addresses the topic in several disciplines. Within the organizational context, employee wellbeing has been related to outcomes such as performance [14] and job satisfaction [27].

Although life satisfaction can help understand relevant future behavior [28], most of the literature has relied on the assumption that work-related feelings and attitudes are relevant to explain nonwork-related feelings and attitudes [16,29]. Assuming a "bottom-up approach" to life satisfaction, these studies conceptualize life satisfaction as the result of satisfaction with life domains. So, what individuals report about their satisfaction with life is a complex and mixed perception of satisfaction with different life domains, such as work, leisure, family and so on [30].

Thus, most of the research on the relation between life satisfaction and work domain has been trying to uncover the role of working behaviors and attitudes on the perceived satisfaction with life. The most common work behavior and attitude is job satisfaction [31], but more recently work engagement has gained considerable attention as a predictor of life satisfaction [32,33].

However, the relation between work-related and nonwork-related feelings and attitudes still deserve attention, since the understanding of mutual relations remains unclear. A study from Judge and Watanabe [34] shows that job satisfaction and life satisfaction were significantly and reciprocally related. Using both cross-sectional and longitudinal approaches, they found that the effects were stronger on the cross-sectional study. The results of the cross-sectional study showed that life satisfaction significantly influenced job satisfaction, and that the effect of job satisfaction on life satisfaction could not be discarded, since the mutual effects are significantly different.

The theoretical support for the relevance of nonwork-related feelings and attitudes in understanding work-related feelings and attitudes could be found in the dispositional perspective [34,35]. According to this perspective, general affective states (in this case, nonwork or life affective states) tend to spill over into evaluations of work-related dimensions, such as feelings and attitudes like work engagement [36–38]. An affective disposition can be understood as the inclination to react to external or environment events in an affect-based manner [35]. In this context, life satisfaction can be understood as the affective disposition that underlies a person's reaction to the environment, in the form of engagement at work.

Thus, the goal of this paper is to contribute to the understanding of the connection between employees' work and personal life, by examining work engagement and life outside work. Specifically, this paper looks at the relation that life satisfaction may have with work engagement.

Although the literature shows that several work behaviors and attitudes, such as job satisfaction and work engagement, are well established as antecedents of life satisfaction, there is almost no research showing that the reverse relation could be supported. One of the few exceptions is recent research that examined the relation of subjective wellbeing to performance, work withdrawal behavior and physical and mental health, when mediated by work engagement [39]. The results supported the assumption that subjective wellbeing was positively related with work engagement. As a consequence, employees' work performance was greater, and work withdrawal behavior and mental and physical ill-health were significantly reduced.

Work engagement lies within the realm of positive organizational behavior [26]. This theoretical approach is based on positive psychology, which emerges from the need to study phenomena and positive psychological outcomes, and how to drive them. Positive organizational behavior looks for "the study and application of positively oriented human resource strengths and psychological capacities that can be measured, developed, and effectively managed for performance improvement in today's workplace" [40] (p. 59). In this context, work engagement—a positive psychological state—can be understood as the antipodes of burnout [41], which is a negative state of mind. More recently, this opposite view of engagement and burnout has been questioned by other supplementary or alternative approaches. For example, Moeller et al. [42] report a study in the US that looked for engagement-burnout profiles. They found that both burnout and engagement could be found in some profiles. Also, meta-analysis research using previous studies on burnout and engagement found insignificant cross-lagged effects between burnout and engagement. However, when time-lag was used as a moderator, burnout and work engagement revealed significant reciprocal effects [43].

Kahn [44] developed one of the first conceptualizations of work engagement as a role-related phenomenon, reflecting the extent to which an individual is psychologically present in a particular organizational role. Following this work, other authors such as Maslach et al. [45], Harter et al. [17], Saks [18] and Czarnowsky [46] have contributed towards the clarification of a work engagement definition. More recently, Schaufeli et al. [47] (p. 74)—a prominent author on the subject—defined work engagement as "a positive, fulfilling, work-related state of mind that is characterized by vigor, dedication and absorption. Rather than a momentary and specific state, engagement refers to a more persistent and pervasive affective-cognitive state that is not focused on any particular object, event, individual or behavior". Vigor refers to levels of energy, mental resilience and persistence. Dedication is about the mental and emotional states that attribute to experience a sense of significance, enthusiasm, inspiration and pride. Finally, absorption refers to the state of being completely concentrated on one's work.

The term 'work engagement' has gained considerable popularity in the past 20 years, yet remains inconsistently defined and conceptualized, with little rigorous academic research done [48]. It is easy to understand the increase in the term's popularity, since the outcomes of work engagement can be exactly what most organizations need. When employees are engaged, they will be more productive and profitable, as well as being less likely to be absent and more willing to work harder for their companies [49,50]. Work engagement can even go beyond the work-specific domain by generating higher customer satisfaction ratings and increased revenue [51].

The measurement of work engagement is a bone of contention among scholars. Saks [18] and Viljevac et al. [52] investigated the validity of two measures of work engagement (the Utrecht Work Engagement Scale (UWES) and the May, Gilson and Harter scale) that have emerged in academic literature. They found some evidence for convergent, discriminant and predictive validity for both scales, although neither showed discriminant validity with regard to job satisfaction. They contend that important differences in measuring engagement raise questions on how to measure the construct, and that the results will be specific to the measures used, limiting generalization.

Following the framework proposed by positive psychology, stressing the need to investigate and find effective applications of positive traits, states and behaviors of employees within organizations, for the purpose of this paper our preferred definition of work engagement is the one provided by Bakker and Schaufeli [26] defining work engagement as a positive, fulfilling, affective-motivational state of work-related wellbeing. This definition presents engagement as an important part of employees' wellbeing in a specific context. As seen previously, life satisfaction is a positive feeling resulting from satisfaction with several dimensions of people's lives [30]. Thus, the inter-relationship established between these different dimensions can anticipate that a general satisfaction with some dimensions of people's lives can also boost their satisfaction with other dimensions. As work engagement is an indicator of work-related wellbeing, it can be expected that there is an association between satisfaction with these dimensions (translated in this context as life satisfaction) and engagement at work.

Based on the assumption that there is a need to test the relation between life satisfaction and work engagement, and also due to the small number of studies that associate work-related attitudes and behaviors with attitudes and perceptions based on the outside-work dimension, this research is based on the following general hypothesis:

**Hypothesis 1:** *General life satisfaction is related to work engagement.*

## 2. Materials and Methods

The aim of this paper is to understand the relation between life satisfaction and work engagement. In order to accomplish this goal, a questionnaire was administered to line managers of several Portuguese medium and large companies. A total of 571 responses were obtained.

Work engagement was measured using the Utrecht Work Engagement Scale (UWES) [41], using a 7-point Likert scale (1 = Not probable; 7 = Most probable) (Appendix A, Table A1). The UWES is one of the most-used constructs to measure engagement. It has been used in several contexts and countries (e.g., [21,22,53–59]). Among these, the study by Schaufeli et al. [41], involving more than 14,000 employees of ten countries, confirmed the UWES construct and validated a short version of the UWES comprising 9 of the initial 17 items. Although smaller, this version kept the initial three dimensions: vigor (e.g., "At my work, I feel bursting with energy"), dedication (e.g., "I am enthusiastic about my job") and absorption (e.g., "I feel happy when I am working intensely").

A few studies report the use of the UWES scale in Portugal. Schaufeli, Martinez et al. [51] used the scale to research students' burnout and engagement, adapting the UWES scale to the goal and population of the study. The scale was translated into Portuguese from the Spanish version, not from the original. Salanova et al. [59] used only two of the three dimensions, meaning they did not use the full version of the scale. Moreover, there is no mention of translation procedures or of the use of Schaufeli's previous translation.

Since neither of the previous studies fit the purpose of this research, we developed our own translation procedures. First, the items were translated to Portuguese and then were subjected to back translation procedures. The results were very good, meaning that the Portuguese translation kept intact the meaning of the original items. Then, we pretested the scale to detect possible misunderstandings.

Life satisfaction was measured using the Satisfaction With Life Scale [60]. This construct measures general life satisfaction and comprises 5 items (e.g., "In most ways my life is close to my ideal"), measured on a 5-point Likert scale (Appendix A, Table A2).

Since some demographic variables may be associated with work engagement [61–63] and life satisfaction, and they are often used as control variables, this study also includes education, tenure, working hours and number of supervised employees as control variables.

The population of this study was made up of employees from some of Portugal's major companies, namely from services (insurances, banking, etc.) and industry (furniture, food and beverages, etc.). The researchers contacted the Human Resources department of each company, which then randomly

distributed the questionnaire by email among their employees. According to the information provided by each company about the employees who received the questionnaire, the rate of response was 11%. The total sample was comprised of 571 employees from several levels and functional areas, as presented in Table 1. Since it was not possible to control respondents, the sample procedure was close to a convenience sample. The criterion to establish the minimum sample size was based on the suggestion of Hair et al. [64] (p. 573–574) for the use of regression analysis, namely the number of constructs and respective number of items, and communalities. Considering these criteria, the minimum number of participants to perform a robust analysis should be 100, which was largely exceeded.

**Table 1.** Sample descriptives.

| Variables | n | % | M | S.D. |
|---|---|---|---|---|
| School degree | | | | |
| Below high school | 9 | 1.6 | | |
| High school | 71 | 12.4 | | |
| Undergraduate | 278 | 48.7 | | |
| Postgraduate | 193 | 33.8 | | |
| PhD | 20 | 3.5 | | |
| Tenure (in years) | | | 11.64 | 7.80 |
| Daily working hours | | | 9.74 | 1.30 |
| Nr. of employees under supervision | | | 34.68 | 93.78 |

Data were collected using a questionnaire with three sections: the first section included sociodemographic questions; the second and third sections included employee engagement and life satisfaction items, respectively. Data analysis included several statistical techniques, namely data reduction and correlation analysis. Factor analysis and reliability analysis were performed on employee engagement and life satisfaction items, allowing researchers to confirm the underlying constructs. After confirming the constructs, mean scores of each factor were calculated. Finally, in order to test the main hypothesis of this research, a hierarchical regression analysis was performed. Several control variables were used, such as school degree, tenure, working hours and number of employees under supervision. Work engagement was computed as the dependent variable and life satisfaction as the independent variable.

## 3. Results

Table 2 shows the mean, standard deviation and correlations for the UWES items. Mean scores are generally high, between 5.69 and 6.27. Most correlations are robust ($r > 0.4$), with only one item presenting a correlation value of $r = 0.378$.

**Table 2.** Mean, standard deviation and correlations for employee engagement items.

| Items | M | S.D. | EE VI1 | EE VI2 | EE DE1 | EE DE2 | EE VI3 | EE AB1 | EE DE3 | EE AB2 | EE AB3 |
|---|---|---|---|---|---|---|---|---|---|---|---|
| EE VI1 | 5.94 | 0.872 | 1 | | | | | | | | |
| EE VI2 | 5.85 | 0.932 | 0.684 | 1 | | | | | | | |
| EE VI3 | 5.69 | 1.198 | 0.524 | 0.666 | 1 | | | | | | |
| EE DE1 | 6.20 | 0.834 | 0.593 | 0.693 | 0.620 | 1 | | | | | |
| EE DE2 | 5.78 | 1.021 | 0.538 | 0.648 | 0.677 | 0.703 | 1 | | | | |
| EE DE3 | 6.27 | 0.192 | 0.468 | 0.557 | 0.571 | 0.637 | 0.674 | 1 | | | |
| EE AB1 | 6.07 | 1.008 | 0.446 | 0.508 | 0.573 | 0.426 | 0.502 | 0.456 | 1 | | |
| EE AB2 | 6.05 | 0.900 | 0.425 | 0.498 | 0.506 | 0.499 | 0.518 | 0.556 | 0.445 | 1 | |
| EE AB3 | 5.88 | 1.129 | 0.378 | 0.423 | 0.469 | 0.438 | 0.483 | 0.425 | 0.475 | 0.529 | 1 |

All *r*-values are significant at p < 0.01.

Life satisfaction items also present relatively high mean scores, but only two items have scores above 5. In terms of correlations, values are quite high with the exception of the item "If I could live my life over, I would change almost nothing" with correlation scores below $r = 0.600$ (Appendix B, Table A3).

Factor analysis was used to reduce data dimensionality for both employee engagement and life satisfaction. Regarding employee engagement, the data were suitable for factor analysis (KMO = 0.923; Barlett Test Sig. = 0.000). Items were grouped in only one component with Percentage of Variance Accounted For of 58.97%. Thus the 9-item UWES scale was adequate, since the one-factor result showed that together, the items reflected the construct of work engagement. The scale presented a robust reliability (UWES9: $\alpha = 0.911$) (Appendix B, Table A4).

Data from life satisfaction were also suitable for factor analysis (Kaiser-Meyer-Olkin = 0.866; Barlett Test Sig. = 0.000). Items were grouped in only one factor, with a total PVAF of 72.35%. Thus the 5-item Life Satisfaction scale seemed to be adequate, since the one-factor result showed that the items reflected the construct. The scale items presented a robust reliability ($\alpha = 0.895$) (Appendix B, Table A5).

In order to test the relation of life satisfaction with general work engagement, a hierarchical regression analysis was conducted. The first block of the regression included education, tenure, working hours, and number of supervised employees as control variables. The second block included work engagement. In order to check the robustness effect, a regression analysis without the control variables was also performed (see Appendix C). Results did not violate the assumptions of collinearity (Tolerance>.10; VIF < 10; $r = 0.406$), normality distribution, outliers or unusual cases. The ANOVA results indicated that the model as a whole was significant [F (5, 555) = 26.50, $p < 0.005$].

The first model, including just the control variables, explained 2.6% of the variance in work engagement. Although low, only "working hours" made a significant and unique contribution ß = 0.107; $p < 0.05$). However, after life satisfaction was entered, Model 2 explained 19.3% of the variance in work engagement. Life satisfaction made a relevant, significant and unique contribution to explaining work engagement (ß = 0.410; $p < 0.05$), followed by "working hours" (ß = 0.108; $p < 0.05$) and "education" (ß = −0.083; $p < 0.05$). The variables "tenure" and "number of supervised employees" made no significant contribution to explaining "work engagement" (Table 3). The results from the regression analysis without the control variables (presented in Appendix C) show similar results, that is, the model explains 19.5% of the variation in work engagement, and life satisfaction made a relevant, significant and unique contribution to explaining work engagement (ß = 0.041; $p < 0.05$).

**Table 3.** Hierarchical regression model.

|  | Independent Variables | $R^2$ | ß | $t$ | Sig. |
|---|---|---|---|---|---|
| Model 1 |  | 0.026 |  |  |  |
|  | Education |  | −0.084 | −1.906 | 0.057 |
|  | Tenure |  | 0.047 | 1.079 | 0.281 |
|  | Working hours |  | 0.107 | 2.521 | 0.012 |
|  | Nr. supervised employees |  | 0.056 | 1.323 | 0.186 |
| Model 2 |  | 0.193 |  |  |  |
|  | Education |  | −0.083 | −2.076 | 0.038 |
|  | Tenure |  | 0.076 | 1.902 | 0.058 |
|  | Working hours |  | 0.108 | 2.785 | 0.006 |
|  | Nr. supervised employees |  | 0.032 | 0.829 | 0.408 |
|  | Life satisfaction |  | 0.410 | 10.700 | 0.000 |

The results showed that life satisfaction had a significant impact on work engagement. This could be verified by the $R^2$ change and life satisfaction's unique contribution. According to Model 2, life satisfaction explained an additional 16.7% of the variance in work engagement, even when the effects of education, tenure, working hours and number of supervised employees were statistically controlled

for. When the overlapping effects of all other variables were statistically removed, life satisfaction made the most significant and unique contribution (ß = 0.410).

## 4. Discussion

When trying to understand what shapes work engagement, the main trend in previous literature has been to look for the reasons for low or high engagement within the limits of the organization. As such, the main factors considered to shape work engagement are mainly organizational, such as job resources and characteristics [21,22], meaningful work [23], organizational support, rewards and recognition and justice [8]. Relying on the employee subjective wellbeing approach, this research paper tried to understand the relation of a broader factor, not confined to the organizational context, such as life satisfaction.

Control variables did not have a significant effect on work engagement. With the exception of working hours, all the remaining control variables were not related to work engagement. The significant positive association of working hours with work engagement may be related to the nature of the sample. A significant proportion of the sample supervised other employees, which was indicative of a management role. Thus, working more hours may have been related to the specific role of those employees. In short, more working hours could be associated with having a management role, which in turn could mean higher levels of engagement [65].

The results show that life satisfaction contributes to explain work engagement. Employees more satisfied with their general life will also show higher levels of engagement with their own work. Contrary to some previous literature, work engagement and life satisfaction can present a different relation [31,32,66]. Nevertheless, there is recent evidence [32,67] that life satisfaction may function as an antecedent of work engagement.

The main conclusion is that work engagement can be related to factors external to the organization, which are an integral part of employees' lives. Being a psychological and emotional state, work engagement can be related to other aspects of employees' lives besides their organizational and job roles. In a way, this assumption was already put forward by previous literature, referring generally to the role that life satisfaction may have on future behavior [28], but also referring to the reciprocity of the life-job satisfaction association [30]. Taking into account the results of this research, but also the strong inclination in the literature to consider work engagement as an antecedent of life satisfaction, it could be also assumed that there is a reciprocal relationship between these two concepts.

Following the assumption that work and personal life domains should not be understood as separate [25], a limitation of this research is the exclusion of work-related variables. Considering work-related variables alongside factors external to the organization could contribute to clarify (and quantify) the importance of life satisfaction, when weighted along other antecedents of work engagement. Thus, considering both work and personal life domains should be considered in future research.

An additional limitation might be the cultural context. The fact that this research was conducted in one specific country (in this case Portugal), with its specific cultural environment, might have conditioned the results. In previous research, Schaufeli [68] demonstrated that work engagement is related to individualistic countries with less power distance and uncertainty avoidance, where the gratification of human needs is valued.

A final limitation is related to the methodological approach. This is a cross-sectional study, which means that it does not capture possible long-term effects of life satisfaction changes on work engagement. Longitudinal studies on this subject should also be conducted in order to understand how far previous perceptions of life satisfaction may be related with future work engagement.

Although some literature supports the impact of work engagement on employees' socio-psychological states, such as happiness [69], this research makes a theoretical contribution by showing that the relation between work and personal employees' lives can run the other way around.

The main empirical contribution of this paper is to call attention to factors outside the sphere of work that might impact on work engagement. This should lead companies to develop more sustainable policies that take into account workers' lives outside work, like Family Friendly Policies [70], not considering work engagement as a simple byproduct of what happens at work. Practitioners should take a holistic approach to addressing employees. This means that when dealing with strategies to improve engagement, practitioners must consider employees' lives outside work, looking for ways to promote a sustainable balance between employees' several roles.

**Author Contributions:** Conceptualization, P.F.; methodology, P.R. and M.S.P.; investigation, S.F. and M.S.P.; formal analysis, P.F.; data curation, C.G.; writing—original draft preparation, S.F. and P.R.; writing—review and editing, P.F. and C.G.; visualization, C.G.; supervision, P.F. All authors have read and agreed to the published version of the manuscript.

**Funding:** This research received no external funding.

**Conflicts of Interest:** The authors declare no conflict of interest.

## Appendix A

**Table A1.** Utrecht Work Engagement Scale items.

| | |
|---|---|
| EE VI1 | At my work I feel bursting with energy |
| EE VI2 | At my job, I feel strong and vigorous |
| EE DE1 | I'm enthusiastic about my job |
| EE DE2 | My job inspires me |
| EE VI3 | When I get up in the morning, I feel like going to work |
| EE AB1 | I feel happy when I am working intensely |
| EE DE3 | I am proud of the work that I do |
| EE AB2 | I am immersed in my work |
| EE AB3 | I get carried away when I'm working |

**Table A2.** Life Satisfaction scale items.

| | |
|---|---|
| LS1 | In most ways my life is close to ideal |
| LS2 | The conditions of my life are excellent |
| LS3 | I am satisfied with my life |
| LS4 | So far I have gotten the things I want in life |
| LS5 | If I could live my life over, I would change almost nothing |

## Appendix B

**Table A3.** Mean, standard deviation and correlations for Life Satisfaction items.

| Items | M | S.D. | LS1 | LS2 | LS3 | LS4 | LS5 |
|---|---|---|---|---|---|---|---|
| LS1 | 4.95 | 1.227 | 1 | | | | |
| LS2 | 4.85 | 1.219 | 0.790 | 1 | | | |
| LS3 | 5.36 | 1.171 | 0.786 | 0.795 | 1 | | |
| LS4 | 5.72 | 1.015 | 0.655 | 0.629 | 0.707 | 1 | |
| LS5 | 4.85 | 1.529 | 0.525 | 0.466 | 0.560 | 0.579 | 1 |

All *r*-values are significant at $p < 0.01$.

**Table A4.** Variance explained, factor loadings and Cronbach's alpha for work engagement.

| Items | % of Variance | Factor Loading | $\alpha$ |
|---|---|---|---|
| WE VI1 | | 0.734 | |
| WE VI2 | | 0.831 | |
| WE VI3 | | 0.819 | |
| EW DE1 | | 0.823 | |
| WE DE2 | 58.97 | 0.841 | 0.911 |
| WE DE3 | | 0.779 | |
| WE AB1 | | 0.691 | |
| WE AB2 | | 0.714 | |
| WE AB3 | | 0.654 | |

**Table A5.** Variance explained, factor loadings and Cronbach's alpha for Life Satisfaction.

| Items | % of Variance | Factor Loading | $\alpha$ |
|---|---|---|---|
| LS1 | | 0.893 | |
| LS2 | | 0.876 | |
| LS3 | 72.35 | 0.914 | 0.895 |
| LS4 | | 0.839 | |
| LS5 | | 0.716 | |

## Appendix C

**Table A6.** Regression model (without control variables).

| | Independent variables | $R^2$ | ß | $t$ | Sig. |
|---|---|---|---|---|---|
| Model 1 | | 0.195 | | | |
| | Life Satisfaction | | 0.041 | 7.711 | 0.000 |

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
