# Peer review of "What if Employees Brought Their Life to Work? The Relation of Life Satisfaction and Work Engagement"

_sustainability, doi:10.3390/su12072743_

Round 1
Reviewer 1 Report
Dear Authors
Please see the attached document to amend the paper.

Author Response
We would like to thank the anonymous reviewers for their insightful comments. In the following points we summarize our reply to the main comments.
- This sentence located in line 270 should be clarified. More specifically the word ‘hole’. “Practitioners should bear in mind to address employees as a hole”.
It was misspelled. It’s now corrected to “whole”
- In the section materials and methods, line 146, the authors mentioned that they obtained 247 responses. Then later in line 178 the sample was 571. Would appreciate if the authors could identify what was the total sample size of this study.
Again, this is typo. The correct sample size is 571. It has been corrected.
- The paper should identify the hypotheses to be addressed in the paper. This is missing in the paper. The hypotheses should be included in the Introduction section and then discussed in the results and discussion sections
The general hypothesis addressed in the research was inserted at the end of the introduction section (line 144).
- The paper lacks theoretical background. It explains the word engagement, but it does not have a theoretical foundation.
We did not explicitly developed the theoretical background since, it is stated in the introduction that “Satisfaction with life is within the realm of subjective wellbeing” (line 64), and a little further “Work engagement lies within the realm of positive organizational behaviour” (line 90). Thus it is implicit that the theoretical background comes from subjective wellbeing and positive organizational behaviour (and psychology).
Reviewer 2 Report
Dear Authors,
Thank you for an opportunity to review this manuscript. Below I list my comments and questions with the hope that it will help you improve on your work.
Abstract:
- Please remove “can be influenced by” as this suggests that you found a causal relationship between life satisfaction and work engagement which is not the case.
- Please provide number of participants in the Abstract.
Introduction Section:
- First part of the Introduction (lines 27-52) is rather lengthy. I understand the need to justify that wellbeing is important part of sustainability at work but this draws attention away from the main topic of your study. Thus, I suggest shortening this section.
- Lines 48-49: you repeat “four dimensions of sustainable HRM”.
- Line 76: please remove/change “influence”: here and throughout the entire manuscript. This suggests causality and your study design doesn’t allow for such conclusions.
- Line 102. Reference on relationship between job burnout and work engagement is rather old. Currently, literature focuses more on profiles of job burnout and work engagement showing that people can experience various combinations of those. See for example: Moeller, Ivcevic, White, Menges, & Brackett, 2018.
Moeller, J., Ivcevic, Z., White, A. E., Menges, J. I., & Brackett, M. A. (2018). Highly Engaged but Burned Out: Intra-Individual Profiles in the US Workforce. Career Development International, 23(1) 86-105. https://doi.org/10.1108/cdi-12-2016-0215
- Lines 119-129: It is not clear to me how discussing the distinction between job and organization engagement contributes to the paper provided that your study is focused on work engagement. I suggest either shortening/eliminating this part or elaborating on it to show the differences between all types of engagement with an emphasis on why you chose work engagement in the end.
- Line 39. I would have liked to see a justification for why Bakker and Schaufeli’s definition was your preference.
- What I find missing in the Introduction is a link between life satisfaction and work engagement. You do state that this is a rarely investigated relationship, however, this paper would greatly benefit from providing rationale for why it is an important one. This is particularly important since, as you mention yourselves, most research focuses on the reversed relationship. Moreover, since the takeaway from your paper is that work engagement can be predicted by factors from outside work domain, I would have liked to see why you think that life satisfaction is the one we should specifically be interested in.
- In summary, I suggest rewriting Introduction and streamline it in a way that provides strong theoretical rationale for life satisfaction as an antecedent of work engagement.
Materials and Methods
- Please provide justification for your sample size. Was it based on power calculation or on feasibility?
- Lines 148-153. I suggest not repeating definition of work engagement. Since you focus on work engagement’s dimensions, they should be described first in Introduction section.
- What is the range for life satisfaction scale? Please use capital letters for the name of the scale.
- Line 174. What do ellipses in parentheses mean?
- In line 178 you state that the sample comprised of 571 employees but in line 146 you say that 247 responses were obtained. What happened to the rest of responses? Please describe the process in details.
- Introducing control variables can change the main result. Thus, each control needs to be carefully justified. Please explain why all these variables were important to control for when testing the relationship between life satisfaction and work engagement. Moreover, because using each of these controls is potentially debatable I would like to see results of regression analysis ran without them (perhaps in a footnote?) to see whether the effect is robust.
- Line 192: I believe you meant to use a different sign (“>” instead of “<”).
- My suggestion is to remove Tables 3, 4 & 5 and keep the results of factor analysis brief. Both UWES and Satisfaction with Life Scale are well-known scales with good psychometric properties. I appreciate that factor analysis is run due to using the scales in a new context, however, this is only preliminary analysis and there seems to be no need for the tables. In particular so because no changes are made to the number of items and scales' reliabilities are good.
- Lines 212-213. Only here we learn that the outcome variable is both work engagement as a whole and its components. This is crucial information that needs to be first mentioned and justified in Introduction.
- Building on the last point, I don’t see the results for components of work engagement as outcomes.
- Line 216: You refer to ANOVA table but there is no table included in the manuscript.
Discussion
- First paragraph in Discussion is a repetition from Introduction. Please consider rewriting.
- “According to this conclusion, work engagement and life satisfaction show a different relation than that of most literature”: this sentence in unclear, please rephrase/elaborate.
- There is no mention of control variables in the discussion of results.
- Limitations should include the fact that the study is cross-sectional and the implication of this.
- In my opinion, interpretation of results is too strong. As I mentioned earlier, all expressions that suggest causation (influence, impact) should be removed from the manuscript as the study design doesn’t allow for them. Moreover, as this is cross-sectional study, no conclusions on time sequence can be drawn.
Author Response
We would like to thank the anonymous reviewers for their insightful comments. In the following points we summarize our reply to the main comments.
- Please remove “can be influenced by” as this suggests that you found a causal relationship between life satisfaction and work engagement which is not the case.
All the phrasing suggesting a casual relationship was replaced by phrases suggesting association.
- Please provide number of participants in the Abstract.
A sentence with the number of participants was added to the abstract.
- First part of the Introduction (lines 27-52) is rather lengthy. I understand the need to justify that wellbeing is important part of sustainability at work but this draws attention away from the main topic of your study. Thus, I suggest shortening this section.
This justification was added as a reply to the Editor’s comments before review. Even though we shorten this section.
- Lines 48-49: you repeat “four dimensions of sustainable HRM”.
This is a typo. It is corrected.
- Line 76: please remove/change “influence”: here and throughout the entire manuscript. This suggests causality and your study design doesn’t allow for such conclusions.
All the phrasing suggesting a casual relationship was replaced by phrases suggesting association.
- Line 102. Reference on relationship between job burnout and work engagement is rather old. Currently, literature focuses more on profiles of job burnout and work engagement showing that people can experience various combinations of those. See for example: Moeller, Ivcevic, White, Menges, & Brackett, 2018.
Moeller, J., Ivcevic, Z., White, A. E., Menges, J. I., & Brackett, M. A. (2018). Highly Engaged but Burned Out: Intra-Individual Profiles in the US Workforce. Career Development International, 23(1) 86-105. https://doi.org/10.1108/cdi-12-2016-0215
The idea was to demonstrate that the two concepts have been accepted as being conceptually distinct for some time; in other words it’s not a new idea. As such, the intention was not to reference the most recent source but one of the most cited and one that approached the topic thoroughly.
We added a new paragraph summarizing more recent approaches to the relationship between the two concepts, including the reference you mention (lines 97-103).
- Lines 119-129: It is not clear to me how discussing the distinction between job and organization engagement contributes to the paper provided that your study is focused on work engagement. I suggest either shortening/eliminating this part or elaborating on it to show the differences between all types of engagement with an emphasis on why you chose work engagement in the end.
This section was deleted.
- Line 39. I would have liked to see a justification for why Bakker and Schaufeli’s definition was your preference.
We added a brief justification for why preferring Bakker and Schaufeli’s definition of work engagement (lines 134-139).
- What I find missing in the Introduction is a link between life satisfaction and work engagement. You do state that this is a rarely investigated relationship, however, this paper would greatly benefit from providing rationale for why it is an important one. This is particularly important since, as you mention yourselves, most research focuses on the reversed relationship. Moreover, since the takeaway from your paper is that work engagement can be predicted by factors from outside work domain, I would have liked to see why you think that life satisfaction is the one we should specifically be interested in. In summary, I suggest rewriting Introduction and streamline it in a way that provides strong theoretical rationale for life satisfaction as an antecedent of work engagement.
In fact, it is a rarely investigated relationship. And in the cases where this relationship is investigated, the assumption is that work engagement plays a role in shaping the perception of life satisfaction. As far as the authors know there is no argument that the reverse case cannot be also true. In other words, there is no research that falsifies this latter assumption, in order to justify following the former argument.
- Please provide justification for your sample size. Was it based on power calculation or on feasibility?
A justification was provided; please see lines 177 through 182.
- Lines 148-153. I suggest not repeating definition of work engagement. Since you focus on work engagement’s dimensions, they should be described first in Introduction section.
The definition was deleted from the Methods section and the characterization of dimensions added to the Introduction (lines 112-115).
- What is the range for life satisfaction scale? Please use capital letters for the name of the scale.
The name of the scale was changed to capital letters and the scale range was specified (lines 168-170)
- Line 174. What do ellipses in parentheses mean?
It’s an example of the items. It was corrected.
- In line 178 you state that the sample comprised of 571 employees but in line 146 you say that 247 responses were obtained. What happened to the rest of responses? Please describe the process in details.
It’s a typo. The correct sample size is 571. It was corrected.
- Introducing control variables can change the main result. Thus, each control needs to be carefully justified. Please explain why all these variables were important to control for when testing the relationship between life satisfaction and work engagement. Moreover, because using each of these controls is potentially debatable I would like to see results of regression analysis ran without them (perhaps in a footnote?) to see whether the effect is robust.
We performed an additional regression analysis without the control variables and present a summary of the results in Appendix C (table C1) (lines 303-304)
- Line 192: I believe you meant to use a different sign (“>” instead of “<”).
Yes, it is now corrected.
- My suggestion is to remove Tables 3, 4 & 5 and keep the results of factor analysis brief. Both UWES and Satisfaction with Life Scale are well-known scales with good psychometric properties. I appreciate that factor analysis is run due to using the scales in a new context, however, this is only preliminary analysis and there seems to be no need for the tables. In particular so because no changes are made to the number of items and scales' reliabilities are good.
Tables 3, 4 & 5 were moved to the Appendix B (298-302).
- Lines 212-213. Only here we learn that the outcome variable is both work engagement as a whole and its components. This is crucial information that needs to be first mentioned and justified in Introduction.
It is a typo. The outcome variable is just work engagement. The reference to the components was removed.
- Building on the last point, I don’t see the results for components of work engagement as outcomes.
Please see previous point.
- Line 216: You refer to ANOVA table but there is no table included in the manuscript.
“The ANOVA table” was replaced by “The ANOVA results” (line 220)
- First paragraph in Discussion is a repetition from Introduction. Please consider rewriting.
The first paragraph of Discussion was re-written (lines 236-242).
- “According to this conclusion, work engagement and life satisfaction show a different relation than that of most literature”: this sentence in unclear, please rephrase/elaborate.
The phrase was re-written (lines 252-253).
- There is no mention of control variables in the discussion of results.
We inserted a new paragraph in the beginning of discussion (lines 243-249), mentioning the lack of significance of the control variables, and also a possible explanation for the significance of working hours.
- Limitations should include the fact that the study is cross-sectional and the implication of this.
This limitation was included in the final part of the Discussion (lines 274-277).
- In my opinion, interpretation of results is too strong. As I mentioned earlier, all expressions that suggest causation (influence, impact) should be removed from the manuscript as the study design doesn’t allow for them. Moreover, as this is cross-sectional study, no conclusions on time sequence can be drawn.
All the phrasing suggesting a casual relationship was replaced by phrases suggestingassociation.
Reviewer 3 Report
Introduction is to long.
Expand discussion.
What is the practical application of the work?
What are the limitation of research?
Expand the references list, more references is out of date.
Author Response
We would like to thank the anonymous reviewers for their insightful comments. In the following table we summarize our reply to the main comments.
- Introduction is to long.
We followed the journal’s guidelines regarding the structure of papers. These guidelines suggest the following structure: introduction, materials and methods, results, discussion (please refer to “Research Manuscript Sections, https://www.mdpi.com/journal/sustainability/instructions).
Thus, the introduction comprises not only the usual topics of an introduction, but also the literature review.
- Expand discussion.
Discussion was partially re-written and expanded.
- What is the practical application of the work?
Please refer to the last paragraph of Discussion (lines 282-288).
- What are the limitation of research?
Please refer to paragraphs 5, 6 and 7 of the Discussion (lines 264-277).
- Expand the references list, more references is out of date.
The reference list contains 62 references: 23% (14) are from the last 5 years and 63% (39) are from the last 10 years. Moreover, in many cases, more recent references do not present new information, making them less relevant.
Even though some references were updated.
Round 2
Reviewer 1 Report
Dear Authors
Thank you for the explanation and the changes made to the paper.
Best wishes
Author Response
Dear Reviewer
Thank you for the time spent reading our paper and for your suggestions.
Best regards
Reviewer 2 Report
Dear Authors,
Thank you for addressing some of the points I raised in a previous round. I believe the manuscript has benefited from these changes. However, I still have a number of comments. My two major concerns remain valid. First, I think your theoretical rationale is lacking and I elaborate on this in my comments below. Second, I am not convinced that you can achieve your goal as I understand it – that is, demonstrating that life satisfaction is antecedent of work engagement – using a cross-sectional design. Even if you do ultimately lay out valid justification for this direction, this is one of these cases where the rationale for reverse relationship can be easily provided (and in fact has been provided) and thus appropriate methodological and statistical methods need to be applied. That being said, your design cannot be changed at this point and therefore below I list those aspects that you still can address.
General:
There are still references to causation. Line 228: “life satisfaction has a significant impact on work engagement”. Also in line 281.
Introduction:
Lines 132-137. I think I can guess your point here but the description is cumbersome and difficult to follow. Perhaps you could try rephrasing.
Regarding your response to the point 9 in my previous review. You are saying in your article that you are interested in a relationship between job satisfaction and work engagement that is reverse to how it usually is studied. Yet in terms of justification you offer that “there is no argument that the reverse case cannot be also true”. That is hardly a theoretical rationale for a hypothesis. Saying that this is a novel relationship is not really true, we already know that these are correlated. In sum, I still don’t see a strong enough justification for your hypothesis.
Results
Thank you for providing sensitivity analysis without control variables but the justification for using them in the first place is still missing. If I understand correctly you added them in response to another Reviewer’s request but the readers won’t know that and will be curious why you included them. Additionally, please also state in the main text whether the result of the analysis without controls was the same as with the controls.
On the same not, in line 242 you say that “As expected, control variables do not have a significant effect on work engagement”. If you didn’t expect them to have an effect on your outcome why include them?
A limitation of cross-sectional study is understated in my opinion. This design not only prevents us from capturing long-term effects but it also doesn’t allow to state that life satisfaction is an antecedent of work engagement, only that these two are related.
Author Response
REPLY TO REVIEWERS’ COMMENTS
We would like to thank the anonymous reviewer for extending this discussion further. In the following text we summarize our reply to the main suggestions.
Reviewers’ Comments
Reviewer 2
- There are still references to causation. Line 228: “life satisfaction has a significant impact on work engagement”. Also in line 281.
We do not find these references on the lines mentioned. Line 228 is the title of table 6; line 281 states “…showing that the relation between work and personal employees’ lives can run the other way around.”
However, we re-wrote the paragraph presenting the main goal of the paper (lines 68-70).
- Lines 132-137. I think I can guess your point here but the description is cumbersome and difficult to follow. Perhaps you could try rephrasing.
The paragraph was re-written.
- Regarding your response to the point 9 in my previous review. You are saying in your article that you are interested in a relationship between job satisfaction and work engagement that is reverse to how it usually is studied. Yet in terms of justification you offer that “there is no argument that the reverse case cannot be also true”. That is hardly a theoretical rationale for a hypothesis. Saying that this is a novel relationship is not really true, we already know that these are correlated. In sum, I still don’t see a strong enough justification for your hypothesis.
a) We are not “interested in a relationship between job satisfaction and work engagement”. We are interested in the relationship between life satisfaction and work engagement.
b) We state and provide support for the idea that factors related with engagement have been searched almost exclusively within the organization;
c) In the few cases where those factors are looked for outside the organization and when specifically involving life satisfaction, we state and provide support for the idea that the focus is mainly on the relevance of work engagement in explaining life satisfaction;
d) Therefore, when we state that “there is no argument that the reverse case cannot be also true” (which, by the way, was stated in the reply to reviewers, not the paper) we are saying that not only the assumption of life satisfaction being relevant to explain work engagement as not been denied, but also that it should be explored in order to be either confirmed or refuted.
e) Although we do not think that this is the context for this discussion, this reasoning follows a Poperian approach to science, namely the empirical falsification.
f) Even though we expanded the theoretical background of life satisfaction, offering arguments and evidence of the work related – non-work related relation discussion (lines 78-92).
- The justification for using control variables is still missing. If I understand correctly you added them in response to another Reviewer’s request but the readers won’t know that and will be curious why you included them.
We understand your point (as you understood ours) and we added a new paragraph in Methods section with some evidence of the use of control variables in similar cases (lines 187-189).
- Additionally, please also state in the main text whether the result of the analysis without controls was the same as with the controls.
We added a paragraph about this on the results (lines 246-249).
- On the same not, in line 242 you say that “As expected, control variables do not have a significant effect on work engagement”. If you didn’t expect them to have an effect on your outcome why include them?
You already answer your own question on point 5 above. We deleted “as expected”.
- A limitation of cross-sectional study is understated in my opinion. This design not only prevents us from capturing long-term effects but it also doesn’t allow to state that life satisfaction is an antecedent of work engagement, only that these two are related.
We’re sorry, but we don’t understand what and if there is a suggestion.
We also modified the title, taking out the expression “antecedent” to be aligned with the content of the article, along with a few minor changes. All new changes are highlighted in yellow.